# Evolutionary Stability of Small Molecular Regulatory Networks That Exhibit Near-Perfect Adaptation

**DOI:** 10.3390/biology12060841

**Published:** 2023-06-09

**Authors:** Rajat Singhania, John J. Tyson

**Affiliations:** 1Graduate Program in Genetics, Bioinformatics and Computational Biology, Virginia Polytechnic Institute and State University, Blacksburg, VA 24061, USA; singhania_rajat@outlook.com; 2Department of Biological Sciences, Virginia Polytechnic Institute and State University, Blacksburg, VA 24061, USA

**Keywords:** perfect adaptation, molecular regulatory networks, evolutionary algorithm, evolutionary stability, incoherent feedforward loops

## Abstract

**Simple Summary:**

If a perfume is released into a room, the occupants will immediately smell it; however, after a few minutes, they will become desensitized to the odor, although it is still in the room, as would be attested by newcomers to the gathering. Such behavior is called ‘adaptation.’ In response to a stepwise increase in an incoming signal (perfume), a sensory cell (the olfactory cell in the nose) sends an output signal (nerve impulses) to an organ (the brain) that responds appropriately (attraction or repulsion, say); however, in the continued presence of the input signal, the sensory cell ceases to respond and reverts to its ‘resting’ state. There are many examples of such near-perfect adaptive responses in the physiology of living cells, and it is natural to inquire as to the underlying molecular bases of such behavior. Using an evolutionary search procedure, this paper examines a wide class of molecular interaction networks for their potential to exhibit near-perfect adaptation. Adaptive networks that are stable to evolutionary fluctuations are characterized by a simple motif with two paths: (i) an incoming signal activates a receptor molecule, which activates an output signal, and simultaneously (ii) the receptor activates a modulator component that inhibits the output.

**Abstract:**

Large-scale protein regulatory networks, such as signal transduction systems, contain small-scale modules (‘motifs’) that carry out specific dynamical functions. Systematic characterization of the properties of small network motifs is therefore of great interest to molecular systems biologists. We simulate a generic model of three-node motifs in search of near-perfect adaptation, the property that a system responds transiently to a change in an environmental signal and then returns near-perfectly to its pre-signal state (even in the continued presence of the signal). Using an evolutionary algorithm, we search the parameter space of these generic motifs for network topologies that score well on a pre-defined measure of near-perfect adaptation. We find many high-scoring parameter sets across a variety of three-node topologies. Of all possibilities, the highest scoring topologies contain incoherent feed-forward loops (IFFLs), and these topologies are evolutionarily stable in the sense that, under ‘macro-mutations’ that alter the topology of a network, the IFFL motif is consistently maintained. Topologies that rely on negative feedback loops with buffering (NFLBs) are also high-scoring; however, they are not evolutionarily stable in the sense that, under macro-mutations, they tend to evolve an IFFL motif and may—or may not—lose the NFLB motif.

## 1. Introduction

Living cells must adapt to environmental conditions in ways that promote their own survival and reproduction (for unicellular organisms) or the fitness of the multicellular organism to which they belong. Cells have evolved sensory systems that detect environmental cues and signal-processing networks that interpret these cues and determine the appropriate response of the cell. In many cases, the appropriate behavior is to detect and respond to an abrupt change in the external signal and then ‘adapt’ (that is, return to the initial stable ‘resting’ state) in the presence of constant stimulus. For example, our sense of smell works this way. We pick up a change in odor in a room but soon become desensitized to the odor. In other words, we go back to the resting state even though the signal (the odor) that triggered the response is still present and will be detected by a new person entering the room. In ‘perfect’ adaptation, the signal-processing network returns to the same resting state regardless of the final, constant level of stimulus (Figure 1A, green line). By this definition, perfect adaptation may be extremely rare, but near-perfect adaptation (Figure 1A, blue line) might be good enough to serve the purposes of a living, responding cell.

Adaptive behaviors are crucial in contexts as varied as chemotaxis in *Escherichia coli* [1,2], chemotaxis [3] and adenylate cyclase activation [4] in *Dictyostelium*, and osmo-response in yeast [5]. Initial mathematical models [6,7] of these behaviors achieved perfect adaptation through fine-tuning of the biochemical parameters, i.e., certain identities among the reaction rate constants must be satisfied in order for the response to return exactly to its pre-stimulus value. In 1997, Barkai and Leibler [8] put forward an alternative explanation of robust perfect adaptation in bacterial chemotaxis based on a detailed model of the interactions among the signal-receptor complex CheA:CheW and its methylase (CheR) and demethylase (CheB). In their model, steady-state receptor activity is independent of signal strength (ligand level), and the biochemical parameters that produce near-perfect adaptation can vary over several orders of magnitude. To get this remarkable result, Barkai & Leibler had to assume that the methylation reaction catalyzed by CheR is always ‘saturated’ (i.e., the rate of the reaction is independent of the concentration of the unmethylated receptor complex), along with three other constraints on the reaction network identified by Yi et al. [9]. As the latter authors remarked, ‘Relaxing any of these four assumptions results in a deviation from exact adaptation.’

Since then, many authors have proposed simple molecular mechanisms (‘motifs’) that achieve robust perfect (or near-perfect) adaptation [9,10,11,12,13,14,15,16,17,18,19,20,21]. We summarize some of this work in Appendix A (A Catalogue of Mechanisms for Robust Perfect Adaptation and Near Perfect Adaptation), and two excellent reviews have been published recently by Ferrell [22] and Khammash [23]. The adaptive motifs fall into three general classes:

Integral Feedback Control. A feedback variable, *Q*(*t*), changes at a rate proportional to the difference between the adaptive response variable, *R*(*t*), and its desired steady-state value, *R*_ss_: $\frac{dQ}{dt}=k\left(R-R_{\mathrm{ss}}\right)$; i.e., $Q\left(t\right)=Q\left(0\right)+k\int_{0}^{t}\left(R\left(\tau \right)-R_{\mathrm{ss}}\right)d\tau$. *Q*(*t*), which measures the deviation of the system from its setpoint, *R*_ss_, then feeds back on the network to cancel the disturbance. If the system comes to a steady state where *Q*(*t*) and *R*(*t*) are no longer changing in time, then $\underset{t\to \infty }{\lim}R\left(t\right)=R_{\mathrm{ss}}$, i.e., perfect adaptation. For examples, see Mechanisms 1, 2, 9, 10, 11 and 14 in the *Catalogue*. In all these cases, the degradation of Q is independent of the concentration of Q, which could be the result of enzymatic degradation, $\frac{dQ}{dt}=-\frac{V_{\max}Q}{K_{M}+Q}\approx -V_{\max}$, in the limit *K*_M_ → 0, as assumed by Barkai and Leibler for the enzyme CheR. Mechanism 14 in the *Catalogue* achieves perfect adaptation by assuming that the feedback variable is synthesized autocatalytically, $\frac{dQ}{dt}=k_{1}QR-k_{2}Q=k_{1}Q\left(R-\frac{k_{2}}{k_{1}}\right)$, so that, at steady-state, $R_{\mathrm{ss}}=\frac{k_{2}}{k_{1}}$ regardless of the incoming signal.Balancing Controls. The signal S upregulates two proteins, P and Q, that have opposite effects on the response variable, R. The activation of R by P is canceled by the inhibition of R by Q. For examples, see Mechanisms 3, 4, 6, 7, 12, and 15 in the *Catalogue*. Mechanism 5 combines balancing and integral feedback controls.Antithetical Feedback. Two components, either P and R or P and Q, bind to make a complex that is removed from the system, thereby canceling the upregulation of R by S. See Mechanisms 8 and 13 in the *Catalogue*.

Two early studies are particularly relevant to our investigations in this paper. In 2008, Francois and Siggia [13] took an ‘evolutionary’ approach to the discovery of motifs exhibiting near-perfect adaptation. Given a small number of interacting proteins, they constructed regulatory motifs randomly from a collection of biochemical interactions (gene expression, proteolysis, phosphorylation/dephosphorylation, and complex formation). Rate-constant values were assigned randomly. One reaction was chosen to receive an external signal, *S*, and one component, R, was chosen as the response variable. The response of the motif to a step change of the signal from *S*_1_ to *S*_2_ at *t* = 0 was simulated, and the motif’s behavior was assessed by a ‘fitness function’ $f=\Delta R_{\max}/\left(\Delta R_{\mathrm{ss}}+\epsilon \right)$, where $\Delta R_{\max}=\underset{t>0}{\max}\left|R\left(t\right)-R_{\mathrm{ss}}\left(S_{1}\right)\right|$ and $\Delta R_{\mathrm{ss}}=\left|R_{\mathrm{ss}}\left(S_{2}\right)-R_{\mathrm{ss}}\left(S_{1}\right)\right|$. Fifty such motifs were simulated simultaneously (in the ‘first generation’), and the 25 highest fitness motifs were chosen to produce a ‘second generation’ of fifty offspring with mutations. Mutations included changes to any kinetic parameter, the creation or removal of regulatory linkages, the reassignment of the response variable, etc. After many generations of mutation and selection, the algorithm settled on two specific motifs (#7 and 8 in the *Catalogue*) for which an increase in *S* induces a transitory increase in *R*(*t*).

In 2009, Ma et al. [14] proposed a different way to search for adaptative motifs. They considered all possible three-node networks of interacting enzymes (e.g., kinases and phosphatases) governed by Michaelis-Menten kinetics:(1)$$\frac{dX_{i}}{dt}=\underset{j}{\sum }\frac{k_{Aj}X_{j}\left(1-X_{i}\right)}{K_{\mathrm{MA}j}+1-X_{i}}-\underset{j}{\sum }\frac{k_{Ij}X_{j}X_{i}}{K_{\mathrm{MI}j}+X_{i}},\;\;i,j=1,2,3$$
where the parameters (*k*_A*j*_, *K*_MA*j*_) and (*k*_I*j*_, *K*_MI*j*_) are, respectively, (k_cat_, Michaelis constant) for enzyme *j* in the ‘activation’ and ‘inactivation’ reactions. X_1_ receives the input signal, X_3_ denotes the adapting response, and X_2_ plays a regulatory role. For each of the ~16,000 motif topologies, they sampled 10,000 randomly chosen parameter values, looking for examples of near-perfect adaptation, as judged by high scores for both precision (roughly ${\left|X_{3\mathrm{ss}}\left(S_{2}\right)-X_{3\mathrm{ss}}\left(S_{1}\right)\right|}^{-1}$) and sensitivity (roughly $\left|X_{3\max}\left(t\right)-X_{3\mathrm{ss}}\left(S_{1}\right)\right|$). An advantage of this approach is that the kinetics of the activation and inactivation reactions can be changed smoothly from ‘linear’ to ‘hyperbolic’ to ‘saturated’ as *K*_M_ is changed from $K_{Mj}\ll X_{i}$ to $K_{Mj}\approx X_{i}$ to $K_{Mj}\gg X_{i}$. Pursuing this approach, Ma et al. [14] identified two ‘minimal’ adaptation networks: motifs 11 and 12 in the *Catalogue*. Notice that both mechanisms rely on saturation kinetics for reactions involving the ‘regulatory component’ (X_3_ in this paragraph, Q in the *Catalogue*). Consequently, motif #11 is an integral-feedback mechanism (because *dQ*/*dt* is independent of *Q*), and motif #12 is a balanced-control network (because *Q*/*P* is independent of *S*).

Later, Shi et al. [17] applied the same approach to three-node networks of interacting transcription factors, $X_{i},\;i=1,2,3,$
(2)$$\frac{dX_{i}}{dt}=k_{Si}\prod \frac{X_{j}^{m}}{K_{j}^{m}+X_{j}^{m}}\prod \frac{L_{k}^{n}}{L_{k}^{n}+X_{k}^{n}}-k_{Di}X_{i},\;\;$$
where the first term describes the synthesis and the second term the degradation of $X_{i}$. In the synthesis term, the first product is over all activators, and the second product is over all inhibitors of transcription. Shi et al. identified two classes of robust perfect adaptation, exemplified by motifs 14 (an integral feedback mechanism) and 15 (a balanced control network) in the *Catalogue*.

## 2. Methods: The Mathematical Model

In this study, we combine the evolutionary approach of Francois and Siggia [13] with the motif-topology framework of Ma et al. [14] and Shi et al. [17]. Following these authors, we consider all possible three-node topologies (Figure 1B), where X_1_ receives the signal, X_3_ generates the response, and X_2_ provides regulatory potential. Instead of the Michaelis-Menten formulation of motif dynamics used by Ma et al. or the Hill function formulation of Shi et al., we model our motifs using ‘Wilson–Cowan’ equations:(3a)$$\frac{dX_{i}}{dt}=\gamma_{i}\left[F\left(W_{i}\right)-X_{i}\right]$$
(3b)$$W_{i}=S\left(t\right)\delta_{i1}+\omega_{i0}+\underset{j\neq i}{\sum }\omega_{ij}X_{j},\;\;\;\;i=1,2,3$$

These dynamical equations were first used by Wilson and Cowan [24] to model excitatory and inhibitory interactions in neural networks. In our context, *X_i_* is the activity of protein *i* (1, 2, or 3), and *F*(*W_i_*) is the instantaneous ‘target’ activity of protein *i*, which depends, through the function *W_i_*, on the net regulation of *X_i_* by the other two variables *X_j_*(*t*), *j* ≠ *i*. At any moment in time, *X_i_*(*t*) is moving towards *F*(*W_i_*) at a rate determined by *γ_i_*. The interaction coefficients *ω_ij_* determine the weights of the ‘influence’ of variable *j* on variable *i* (*ω_ij_* > 0 for activation, <0 for inhibition, and =0 for no interaction). The offset *ω_i_*_0_ determines *W_i_* in the absence of any regulatory influences on node *i*. For node 1, a signal term *S*(*t*) is added to *W*_1_. For *F*(*W_i_*) we choose the hyperbolic tangent function (‘soft Heaviside’ function):(3c)$$F\left(W_{i}\right)=\frac{1}{2}\left[1+\tanh\left(\frac{\sigma W_{i}}{2}\right)\right]=\frac{1}{1+e^{-\sigma W_{i}}}$$
The parameter *σ* determines the steepness of the sigmoidal curve (Figure 2A). Note that 0 < *F*(*W_i_*) < 1; hence, 0 < *X_i_*(*t*) < 1 for all *t* ≥ 0. In all our calculations, we choose *σ* = 10.

Equation (3a–c) have recently been used advantageously to model protein- and gene-regulatory networks [25,26,27,28]. For our purposes, the Wilson–Cowan equations have similarities to Equation (2) of Shi et al. [17] and have certain advantages over Equation (1) of Ma et al. [14]. Equation (3) provides a simple and more flexible formalism for modeling interaction networks of any complexity. In practical terms, it is easy for us to introduce both micro- and macro-mutations into the model simply by changing parameters (the interaction coefficients *ω_ij_*) without changing the form of the differential equations. Micro-mutations correspond to changes in the value of an *ω_ij_* without changing its sign; macro-mutations correspond to a change in the sign of an *ω_ij_*. In conceptual terms, the Wilson-Cowan approach is closely related to the well-studied piecewise-linear approximation to Boolean dynamical systems [29], because $\underset{\sigma \to \infty }{\lim}F\left(W\right)=\mathrm{Heav}\left(W\right)$ = 0 if *W* < 0 and = 1 if *W* ≥ 0. This recognition brings the intuitive appeal of Boolean dynamics into our understanding of the dynamical behavior of Equation (3a–c).

Our goal is to identify networks of three interacting proteins (and/or genes) that exhibit near-perfect adaptation to a stepwise increase in signal (*S* = 0 to *S* = 1). In principle, this involves searching a 16-dimensional parameter space P = {*ω_ij_*, *ω_i_*_0_, *γ_i_*, *σ*}, subject to the bounds specified in Table 1. To achieve our goal, we plan: (1) to introduce some simplifying assumptions; (2) to subdivide the parameter space into subsets associated with three-link networks (called *pure* signaling motifs); and (3) to pursue an evolutionary search strategy similar to Francois and Siggia [13].

First, the search strategy. We create a population of potential adaptive networks (drawn from the parameter space P) and assess each one’s ‘fitness’ according to the ‘scoring’ function:(4)$$Z=\Delta X_{3\max}/\left(\Delta X_{3\mathrm{ss}}+0.05\right)$$
where $\Delta X_{3\max}=\underset{t>0}{\max}\left|X_{3}\left(t\right)-X_{3\mathrm{ss}}\left(S_{1}\right)\right|$ and $\Delta X_{3\mathrm{ss}}=\left|X_{3\mathrm{ss}}\left(S_{2}\right)-X_{3\mathrm{ss}}\left(S_{1}\right)\right|$. For an example of the fitness function, see Figure 2B. (We shall refer to *Z* interchangeably as the ‘fitness’ or the ‘score’ of a network.) Based on their fitness, networks are chosen to contribute progeny to the next generation. The population of competing networks is then allowed to evolve from one generation to the next to achieve the highest possible score. (More details on the evolutionary algorithm are provided in Appendix B). In creating progeny, we allow for both micro-mutations (a fixed sign pattern of the *ω_ij_*’s but random changes to parameter values) in order to identify regions of parameter space (if any) where a particular network topology can exhibit near perfect adaptation and macro-mutations (allowing network topologies to mutate as well) to investigate the evolutionary relationships among competing patterns of network interactions.

Second, we make two simplifying assumptions. (i) We do not allow for self-regulation, i.e., *ω*_11_ = *ω*_22_ = *ω*_33_ = 0. This assumption greatly reduces the number of distinct topologies from 3^9^ = 19,683 to 3^6^ = 729. It is reasonable because most earlier studies of near-perfect adaptation (refer to the *Catalogue*; motif #14 is the exception) did not find any significant role for self-activation or self-inhibition. (ii) In implementing our evolutionary algorithm, we require that, for *S*_2_ > *S*_1_, $\underset{t>0}{\max}X_{3}\left(t\right)>X_{3\mathrm{ss}}\left(S_{1}\right)$, i.e., that an increase in signal induces an initial rising response (i.e., a ‘bump’ rather than a ‘dip’). This restriction reduces the complexity of the search process and the interpretation of the results. Furthermore, when considering the evolutionary stability of networks, we suppose that a macro-mutation that converts a bump response into a dip response will be disadvantageous because the response is going in the opposite direction. So, we assume that fitness = 0 for all dip-responses. Once we understand the evolutionary relationships of bump-responders, it is possible to convert them to dip-responders by changing the signs of particular interaction coefficients.

Third, we subdivide the problem into smaller domains of parameter space by exploiting the notion of ‘minimal’ signaling topologies, as explained in the next section.

## 3. Results

### 3.1. Classifying ‘Minimal’ Topologies That Might Exhibit Near-Perfect Adaption

Ma et al. [14] introduced the concept of ‘minimal’ three-node topologies that have exactly three non-zero interaction coefficients. Within our model framework, it is trivial to identify any network topology by its unique ‘sign pattern’ (*s*_12_, *s*_13_, *s*_21_, *s*_23_, *s*_31_, *s*_32_), where *s_ij_* = sign(*ω_ij_*) = +1 if *j* activates *i*, −1 if *j* inhibits *i*, and 0 if there is no influence. A minimal three-node topology corresponds to a sign pattern with three zero and three non-zero entries; hence, there are 6-choose-3 = 20 different sign patterns (i.e., minimal three-node, three-link topologies), which are displayed in Appendix A. Each sign pattern has 2^3^ = 8 distinct cases, corresponding to choosing either +1 or −1 for the non-zero entries. Hence, among the 3^6^ = 729 possible topologies, there are 160 minimal signaling motifs. Not all of these motifs are candidates for near-perfect adaptation. In fact, only the first five sign patterns in Appendix A are likely candidates; the other fifteen are unlikely for the reasons specified in the table. As a first step in identifying adaptive motifs, we allow a partially random collection of network topologies to evolve together (with micro- and macro-mutations) in order to see which topologies come out on top.

First, we introduce a convenient ‘code’ for network topologies: (*d*_12_, *d*_13_, *d*_21_, *d*_23_, *d*_31_, *d*_32_), where *d_ij_* = *s_ij_* + 2. With this change, we replace a sign pattern with a digital code consisting of the integers 1 (inactivation), 2 (no influence), and 3 (activation).

### 3.2. Initial Exploration of Topology Space

As a first step, we explored topology space by picking forty starting topologies (generation 0) and allowing them to compete (based on their fitness) and reproduce (with mutations) from one generation to the next. Twenty-four of the forty starting topologies were generated by randomly selecting each of the six digits of the topology’s code, thereby ensuring that a variety of sign patterns (i.e., network topologies) were given a chance to exhibit adaptation. The other sixteen topologies were chosen from four categories: incoherent feed-forward loops (IFFLs), coherent feed-forward loops (CFFLs), negative feedback loops with buffer nodes (NFLBs), and positive feedback loops with buffer nodes (PFLBs). Each of these categories was represented by four different cases. For each starting topology, we chose a random point in parameter space consistent with the topology’s code and the bounds in Table 1. In each generation, every topology suffered micro-mutations by making small random changes to *ω_ij_* values (without changing the topology’s code), and some topologies suffered macro-mutations by changing their code (i.e., changing one *d_ij_*) and then assigning the value of *ω_ij_* from the appropriate range (Table 1). For more details, see Section A.3: *Evolutionary Algorithm*.

Collectively, more than 500 unique topologies were generated across the forty runs. A particular topology may be present in more than one run, so the outputs from all forty runs were collated before calculating the average score of each topology over its lifetime. The scores ranged from nearly 0 to ~14. All topologies that scored >6 are listed in Table 2. Our first observation is that all top-scoring topologies contain incoherent feed-forward loops. There are four distinct IFFLs, identified in Figure 3A. Of the 28 top-scoring topologies, 20 carry an IFFL-1 motif (XX3X31) and 8 carry an IFFL-4 motif (XX1X33). (Not surprisingly, IFFL-2 and IFFL-3 motifs are missing because they are associated with dip-responses rather than bump-responses.) Mixed in among the IFFL-1 and -4 topologies (red and green) in Table 2 are NFLB-1, -2, -3, and -4 motifs (Figure 3B). Surprisingly, our initial search did not find any high-scoring NFLB-1 or NFLB-3 topologies uncoupled from IFFLs, even though Ma et al. [14] identified NFLB-1 and NFLB-3 as minimal adaptation networks. This observation prompted us to look more closely at how adapting networks evolve under micro-mutations only and then under macro-mutations as well.

### 3.3. Close Examination of IFFL and NFLB Topologies

Since Ma et al. [14] identified two classes of ‘minimal’ motifs (we call them *pure* motifs) that exhibit near-perfect adaptation, namely IFFLs and NFLBs, we examined these two classes separately for near-perfect adaptation. These motifs are diagramed in Figure 3A,B and identified by ‘name’ and ‘code’ (*d*_12_*d*_13_*d*_21_*d*_23_*d*_31_*d*_32_).

#### 3.3.1. IFFL Topologies

The code for each *pure* IFFL motif in Figure 3A has three 2s that can take any value 1, 2, or 3 without breaking the IFFL, so there are 27 members of each IFFL category that need to be examined. Recall that we are only looking for bump responses (not dip responses), so we focus on IFFL-1 and -4. We start by examining all 27 IFFL-1 and all 27 IFFL-4 motifs for their ability to evolve near-perfect adaptation under micro-mutations only (i.e., changes in the values of the *ω_ij_*’s without changing the code of the motif; see the methods subsection). Each simulation starts with a set of randomly chosen parameter values that yield a low score. For each generation, we compute the average score of the parental parameter sets. In Figure 4, we plot some examples of how these average scores change from generation to generation.

For each simulation, we monitor the first passage time (FPT), i.e., the number of generations it takes for the average score to exceed 10, and stop the simulation 50 generations after its FPT. Then we calculate the topology’s overall average score over the last 50 generations; see Table 3. All 27 IFFL-1s and 27 IFFL-4s achieve high-scoring results even from a poor start, showing that these two classes of topologies can exhibit near-perfect adaptation. A similar search of IFFL-2 and IFFL-3 topologies found much lower average scores (not shown), as expected.

These simulations were done with *N* = 20 parents per generation and *R* = 20 offspring per parent. With fewer than 20 progeny per parent, the evolutionary algorithm often does not find a high-scoring region.

#### 3.3.2. NFLB Topologies

Next, we carried out the same micro-mutation-only simulations of the four NFLB cases (Figure 3B), and we summarize the results of all 4 × 27 topologies in Table 4. The highest-scoring topologies are (with one exception) NFLBs combined with IFFL-1 or IFFL-4. Later, we shall examine the relative contributions of IFFLs and NFLBs to these topologies, but first we investigate NFLBs that are not coupled to IFFL-1 or -4 (black in Table 4), referred to as *uncoupled* NFLBs.

Starting from the four *pure* NFLBs shown in Figure 3B, we find that only the *upper* NFLBs (NFLB-1 and NFLB-2) score decently on their own (9.80 and 7.10, respectively), in contrast to the *lower* NFLBs (-3 and -4), which score poorly on their own (3.13 and 2.64). These results contradict the conclusions of Ma et al. [14] that NFLB-1 and NFLB-3 are ‘minimal’ adaptive networks, and NFLB-2 and -4 are not. In general, the *uncoupled* NFLB-1 topologies score highest (8.6–15.4), followed well behind by *uncoupled* NFLB-2s (≤8.2) and *uncoupled* NFLB-3s (≤5.7). With three exceptions, *uncoupled* NFLB-4s score poorly (≤3.2).

In summary, NFLB-1 and (to a lesser extent) NFLB-2 topologies score well on their own and particularly well when combined with IFFL-1 and IFFL-4 topologies, respectively. NFLB-3 and NFLB-4 (with three exceptions) score well only when combined with IFFL-1 and IFFL-4, respectively.

#### 3.3.3. Evolution of IFFL-1 and IFFL-4 Topologies under Macro-Mutations

Next, we allowed the high-scoring IFFL-1 or IFFL-4 topologies to macro-mutate. Starting from the highest-scoring parameter set obtained from the micro-mutations only run, the IFFL-1 topologies drift among themselves, keeping their high scores (e.g., Figure 5), and similarly for the IFFL-4 topologies (not shown). These results suggest that the IFFL-1 and IFFL-4 topologies (the two sets of 27 motifs in Table 1) form two separate ‘mesas’ in topology space, in the sense that (1) they both exhibit very high average scores when examined on their own; and (2) they are evolutionarily stable, i.e., they stay within themselves when allowed to macro-mutate.

First, let us define a ‘mesa’ in topology space T = {(*d*_12_, *d*_13_, *d*_21_, *d*_23_, *d*_31_, *d*_32_) | *d_ij_* = 1, 2, or 3}. The ‘mesa’ for a *pure* motif, e.g., IFFL-1, is a subset M_IFFL1_ = {(XX3X31) | each X = 1, 2, or 3}. Each mesa of a *pure* motif has 3^3^ = 27 elements. Since there are 160 minimal motifs and each one has a mesa of 27 elements, there is a great deal of overlap among the mesas. Of the 160 *pure* motifs, most of them are unconducive to bump responses, so we only need to focus on the subsets of those *pure* motifs that show up as ‘adaptive’ in our evolutionary searches.

#### 3.3.4. Why Are IFFL-1 Topologies Evolutionarily Stable?

To explore the evolutionary stability of the IFFL-1 motif in topology space, we focus on the region of T occupied by the mesas M_IFFL1_, M_NFLB1,_ and M_NFLB3_. These subsets overlap, as indicated in Figure 6A. The union of the three mesas, called U, contains 63 topologies, and we arrange these topologies on a Venn diagram on the basis of their average scores, < Z >, recorded in Table 3 and Table 4. We stratify U with dashed lines of fixed *Z* in intervals of 2 units, from the highest score of 18 to the lowest of 2. In the absence of macro-mutations, each topology will climb from a random location in the desert region of parameter space to the peak fitness for that topology recorded in Figure 6A.

Each topology can undergo 12 different macro-mutations that flip a single *d_ij_*. For example, in Figure 6A, we show how the *pure* IFFL-1 topology (223231) can macro-mutate to six other topologies in U (the solid red arrows induced by changing any ‘2′ to a ‘1′ or ‘3′) or to six topologies outside U (the dashed red arrow pointing into the desert region). For the six macro-mutations that end up within U, further macro-mutations will tend to drive the network to the high-scoring region, say *Z* > 16. In particular, any mutant topology that starts in M_IFFL1_ will remain in M_IFFL1_, i.e., IFFL-1 topologies are evolutionarily stable. On the other hand, a mutant topology that starts in M_NFLB3_ but not in M_IFFL1_ will clearly evolve up the fitness landscape into M_IFFL1_. So M_NFLB3_ is not evolutionarily stable. Similarly, although the fitness gradient is not so steep, a topology that starts in M_NFLB1_\M_IFFL1_, can be expected to evolve by macro-mutations into M_NFLB1_ ∪ M_IFFL1_. We have confirmed this expectation by numerical simulations: in Appendix A, we trace the fate of the 18 *uncoupled* NFLB-1 topologies, showing that they evolve by macro-mutations predominantly (>99%) into IFFL-1 + NFLB-1 topologies (1X3X31), which are the highest fitness topologies, according to Figure 6A. (Two of the simulations did not reach a high-scoring endpoint.) Appendix A shows that *uncoupled* NFLB-3 topologies macro-mutate predominantly into IFFL-1 + NFLB-1 topologies (1X3X31), as well. In one case (212331), the final state was predominantly IFFL-1 + NFLB-3 or IFFL-1 only.

In Figure 6B, we show the Venn diagram of U′ = M_IFFL4_ ∪ M_NFLB2_ ∪ M_NFLB4_, which lies in a different region of topology space. The diagram confirms that M_IFFL4_ is evolutionarily stable, but M_NFLB2_ and M_NFLB4_ are not. Surprisingly, when NFLB-2 and NFLB-4 topologies are subjected to macro-mutations, they tend to evolve into IFFL-1 + NFLB-1 topologies (see Appendix A). To understand why, consider that many macro-mutations kick a starting topology out of both U and U′ into the desert region, where mutants are likely to go extinct or, by some chance, find their way back into U or U′. In particular, it is impossible to move between M_IFFL1_ and M_IFFL4_ (the evolutionarily stable mesas) by a single macro-mutation. For example, three of the six macro-mutations that move the *pure* IFFL-1 motif into the desert are coherent feed-forward loops (CFFLs): 221231, 223233, and 223211. Any of them can revert back to IFFL-1 (223231) by a second macro-mutation; two of them can mutate further to IFFL-4 (221233); the third can mutate to IFFL-2 (221211) or IFFL-3 (223213), which are not bump-responders and, hence, are weeded out by selection. Since the fitness of topologies in U is generally higher than that in U′, it is reasonable to expect that a sequence of macro-mutations that push mutants around in the desert region is more likely to end up in U than in U′, as we observe.

#### 3.3.5. Examining the Interactions of IFFL and NFLB Topologies

Figure 3C shows the four-link topologies that result from coupling the IFFL and NFLB topologies. Table 4 shows the average scores of the four NFLBs coupled with IFFL-1 (red) and IFFL-4 (green). In almost all cases, these topologies have higher average scores than the *uncoupled* NFLBs (black). To quantify the overall increase in score of an NFLB motif by the addition of an IFFL, we build tables that show the percentage change in average score per NFLB topology. For example, Appendix A records the percentage increase in score of an *uncoupled* NFLB-1 topology (an element of M_NFLB1_\M_IFFL1_) when the uncoupled topology is mutated to include an IFFL-1 motif. The change in score is averaged over all *uncoupled* NFLB-1 topologies to get an overall percentage change (+52%), as shown in Figure 7A, leftmost panel. Appendix A correspond to the other panels in Figure 7A.

Next, we check the effect of adding NFLBs to IFFLs; the results are recorded in Appendix A. From these tables, we calculate the average change in score across all combinations and summarize the results in Figure 7B. Similar results for adding NFLB-2 and/or NFLB-4 to IFFL-4 only are recorded in Appendix A and summarized in Figure 7C. In all cases, adding an NFLB increases an IFFL score by a much lower percentage than when IFFLs are added to an NFLB. Therefore, we conclude that IFFLs contribute much more significantly to the coupled topologies’ scores than do NFLBs.

Interestingly, adding NFLB-1 (an upper NFLB) increases IFFL-1 scores, whereas adding NFLB-3 (a lower NFLB) decreases the scores. This pattern is reinforced by Figure 7C, where adding the upper NFLB-2 increases the fitness of IFFL-4 much more than adding the lower NFLB-4. These findings are consistent with our earlier observation that the uncoupled upper NFLBs tend to score better on their own than the uncoupled lower NFLBs. In the next section, we validate the contributions of upper NFLBs to high-scoring IFFLs, as compared to lower NFLBs.

#### 3.3.6. Interaction Coefficients Measure the Relative Contributions of NFLB and IFFL Motifs to High-Scoring Combination Topologies

To check that upper NFLBs contribute more significantly than lower NFLBs to high-scoring NFLB+IFFL combinations, we compare the means of all the relevant interaction coefficients (the *ω_ij_*’s) in the combination topologies. For each of the micro-mutation-only simulations presented earlier, we identify the high-scoring topologies (score ≥ 10) and calculate the means of all six *ω_ij_*’s. Appendix A records their values for IFFL-1 topologies.

From these results, we first observe that the three *ω_ij_*’s corresponding to the underlying IFFL-1 class (i.e., *ω*_21_, *ω*_31,_ and *ω*_32_) have weights close to their most extreme values of ±1. The interaction strengths are also strong in the case of negative *ω*_12_′s, which account for the negative feedback loop of the upper NFLB-1. In the case of positive *ω*_23_′s, which account for the negative feedback loop of the lower NFLB-3, the interaction coefficients are weak (close to their minimum possible value of 0.1). This is clear evidence that the IFFL-1′s, along with the upper NFLB-11’s, are driving the near-perfect adaptive responses. The lower NFLB-3s play negligible roles in adaptation.

We reach a similar conclusion for IFFL-4s (see Appendix A). The interaction strengths are strong in almost all cases of positive *ω*_12_′s, which account for the negative feedback loop of the upper NFLB-2. The negative *ω*_23_′s, which account for the negative feedback loop of the lower NFLB-4, are weak (close to their weakest possible value of −0.1). Clearly, it is IFFL-4′s in combination with upper NFLB-2′s—not lower NFLB-44’s—that are driving the near-perfect adaptive responses.

These results show that IFFLs combined with upper NFLBs score best among all classes, and the interaction coefficients for these motifs tend to be close to ±1 in the high-scoring sets.

Appendix A record the means and standard deviations of all model parameters for IFFL-1 and -4 topologies, respectively.

## 4. Discussion

### 4.1. Summary of Results

Three-component signaling motifs (see Appendix A) are common in large-scale regulatory networks involving interacting enzymes and transcription factors [30,31]. For example, in the transcription network of *Escherichia coli*, Ma et al. [32] have identified ~700 feed-forward loops, of which 70% are CFFLs and 30% are IFFLs (mostly CFFL-1 and IFFL-1). As many authors have shown, these motifs play specific functional roles in the physiology of living cells [31,32,33,34,35]. Of special interest here is the ability of cells to respond to an abrupt stimulus but subsequently ‘adapt’ to a sustained stimulus by returning to the pre-stimulus ‘resting’ state. Small network motifs are known to facilitate adaptive responses in a variety of circumstances. For example, Takeda et al. [3] have attributed near-perfect adaptation in the Ras signaling pathway in *Dictyostelium* amoebae to an IFFL motif; Muzzey et al. [5] have shown that osmoregulation in yeast cells is governed by integral feedback control; Basu et al. [36] have engineered *E. coli* cells to exhibit pulse-like expression of GFP in response to an inducer, acyl-homoserine lactone; Csikasz-Nagy et al. [37] have implicated IFFLs in the initiation of DNA replication and mitotic exit during reproduction of yeast cells; and O’Donnell et al. [38] have implicated an IFFL in the regulation of E2F1 gene transcription by c-Myc and micro-RNAs.

In this work, we propose a mathematical model for studying the evolutionary stability of near-perfect adaptation in simple three-component networks of interacting genes and/or proteins. To this end, we model the dynamical behavior of these networks in terms of a standardized, flexible set of ordinary differential equations, Equation (3a–c), introduced years ago by Wilson and Cowan [24] to model neural networks and used more recently by numerous authors to model gene/protein interaction networks [25,26,27,28]. A Wilson–Cowan network on components X_1_, X_2_, and X_3_ is parameterized by a set of interaction coefficients (*ω_ij_*, *i,j* = 1,2,3) whose signs specify the topology of the network (*ω_ij_* > 0 for activation, <0 for inhibition, and = 0 for no interaction), and whose magnitudes determine the strength of the interaction. The Wilson–Cowan approach is especially useful for us because we can easily model mutations in the signaling network by changing one interaction coefficient at a time. Micro-mutations correspond to changing the magnitude of an *ω_ij_* without changing its sign, and macro-mutations correspond to changing the sign (+, − or 0) of an *ω_ij_*. We follow the evolution of the network over the course of many generations by introducing, for each specific ‘parental’ network, a ‘fitness’ function, *Z*, that quantifies its behavior as a near-perfect adaptive response to a stepwise increase in signal. Parental networks contribute ‘offspring’ to the next generation in accordance with their fitness. We have followed the evolution of the signaling networks under these conditions for many generations to identify winners and losers. Since we are interested in the evolutionary stability of networks, we must penalize disadvantageous mutations that convert a ‘bump’ response (a transient increase) to the signal into a ‘dip’ response (a transient decrease); therefore, our fitness function only rewards bump responses. In addition, we limit ourselves to networks without self-activation or inhibition, i.e., *ω_ii_* = 0 for *i* = 1,2,3. In this case, our space of possible network topologies, T, has 3^6^ = 729 elements.

We summarize the results of our study schematically in Figure 8. In the space T, we find two separate ‘mesas’ of high fitness towering above a vast ‘desert’ of non-adaptive networks. The mesas are associated with incoherent feed-forward loops of types 1 and 4 (IFFL-1 and IFFL-4; see Figure 3A). Each mesa has two shoulders associated with negative feedback loops with a buffering node (NFLB-1, -2, -3, -4; see Figure 3B). Each mesa, with its shoulders, accounts for 63 different topologies; so, the desert accounts for 603 non-adaptive topologies. In set notation, the ‘IFFL-1 mesa’ is U = M_IFFL1_ ∪ M_NFLB1_ ∪ M_NFLB3_, where M_IFFL1_ is the subset of 27 topologies that contain an IFFL-1 sign pattern, and similarly for M_NFLB1_ and M_NFLB3_. We denote the ‘IFFL-4′ mesa as U′ = M_IFFL4_ ∪ M_NFLB2_ ∪ M_NFLB4_. If we consider macro-mutations that change one topology into another, we find that macro-mutations that stay within the ‘IFFL-1 mesa’ evolve to the top of the mesa, so the subset M_IFFL1_ is evolutionarily stable, but the shoulders, M_NFLB1_ and M_NFLB3_, are not because they may lose the negative feedback loop during the course of evolution. Similarly, M_IFFL4_ is evolutionarily stable, but the shoulders, M_NFLB2_ and M_NFLB4_, are not.

Some macro-mutations carry a topology out of U (or out of U′) into the desert. These mutants will have very low fitness and are likely to go extinct unless they experience a reverse mutation back into U (or back into U′) or a further macro-mutation taking them into U′ (or into U). In this sense, M_IFFL1_ ∪ M_IFFL4_ is evolutionarily stable with respect to all mutations. Since the IFFL-1 mesa exhibits higher fitness scores than the IFFL-4 mesa, we might expect IFFL-1 topologies to dominate over IFFL-4 topologies over the long run of evolutionary history. In fact, IFFL-1 motifs are ~ten times more prevalent than IFFL-4 motifs in modern transcriptional networks in *E. coli* and budding yeast [33,34,35]. Alon has given some reasons why IFFL-4 motifs are (unexpectedly for us) rare in adaptive signaling networks; see Section 4.8 of [31].

### 4.2. Comparison with the Results of Ma et al. and Shi et al.

Our approach differs from that of Ma et al. [14] and Shi et al. [17] in several respects (see Appendix C), and our results extend their work in several directions. Not only have we quantified more precisely the dominance of IFFLs over NFLBs, but we have also shown which particular classes of IFFLs and NFLBs score best. In fact, one of our adaptive motifs (IFFL-4) was identified by Ma et al. as non-adaptive; see their Figure 2B. We show clearly that both IFFL-1 and IFFL-4 exhibit near-perfect adaptation to a very high degree, forming two separate ‘mesas’ in topology space. Since we are looking only for bump responses, IFFL-2 and IFFL-3 topologies have low fitness. Of course, they would both do well in a search for dip-responses.

An important mathematical difference between our model and Ma et al.’s needs to be highlighted. Since they consider only small changes in input (from 0.5 to 0.6), they are able to linearize the underlying Michaelis–Menten ODEs, and from their linearized equations, they show that, for NFLBs to show perfect adaptation, $J_{22}^{0}≅0$, where $J_{22}^{0}$ is a diagonal element of the Jacobian matrix of the system at steady state. $J_{22}^{0}≅0$ is satisfied when the enzymes acting on Node 2 (our variable X_2_) are in saturation, i.e., when the ODE for Node 2 has Michaelis constants much smaller than substrate concentrations. Under this condition, Node 2 implements integral feedback control in NFLBs by integrating the difference between the activity of response Node 3 and Node 3′s signal-independent steady state value. See Equations (2)–(4) in Ma et al. [14] for the mathematical details. In our model, however, there is no requirement for integral feedback control ($J_{22}^{0}≅0$) to achieve adaptation in NFLBs; indeed, from Equation (3a), we find $J_{22}^{0}=-\gamma_{2}$, which is small but not zero (i.e., 0 < *γ*_2_ < *γ*_3_ ≡ 1 < *γ*_1_).

In Appendix D, we discuss the ‘fine-tuning’ of parameter values in relation to our model and Ma et al.’s.

Our work bears more similarity to Shi et al. [17] because the Wilson–Cowan equations we use are applicable to both enzymatic regulatory networks and transcriptional regulatory networks [26]. However, we do not find their case of a negative feedback loop with exponential buffering (Mechanism #14 in the *Catalogue*) for two reasons: (1) we have excluded self-activation in the network, and (2) even if we were to allow the term $\omega_{22}X_{2}$ in Equation (3b), we would not reproduce Shi’s result because the autocatalytic effect of the regulatory node ($X_{2}$) is highly nonlinear in the Wilson–Cowan equations.

## 5. Conclusions

In summary, using an evolutionary search algorithm similar to [13] and a modeling formalism similar to [24], we have extended the work of Ma et al. [14] and Shi et al. [17] by identifying the specific combination of IFFLs and NFLBs that are most conducive to near-perfect adaptation. We confirmed the evolutionary stability of the optimal topology classes, a result beyond the scope of the methodology used by other investigators. Finally, we have provided a more concrete picture of the regions in parameter space where the high-scoring topologies exhibit near-perfect adaptation.

None of the models of perfect and near-perfect adaptation in the *Catalogue* (Appendix A) consider the potential effects of molecular fluctuations that are inevitable in signaling networks in bacterial and yeast cells, which are small in volume and have limited numbers of macromolecules (maybe only 5–10 copies of crucial mRNAs). Characterizing these effects is an important issue for future study.

Our evolutionary approach can be used to find motifs and parameter sets that display other behaviors, such as bistability, oscillations, Turing patterns, and chaos. Once the appropriate scoring function has been designed in each case, the evolutionary approach is the same as the one taken here. By confirming that our approach works for the case of near-perfect adaptation, we have taken the first step in creating a topological structure-function map that could be a useful tool for systems biologists.

## Figures and Tables

**Figure 1 biology-12-00841-f001:**
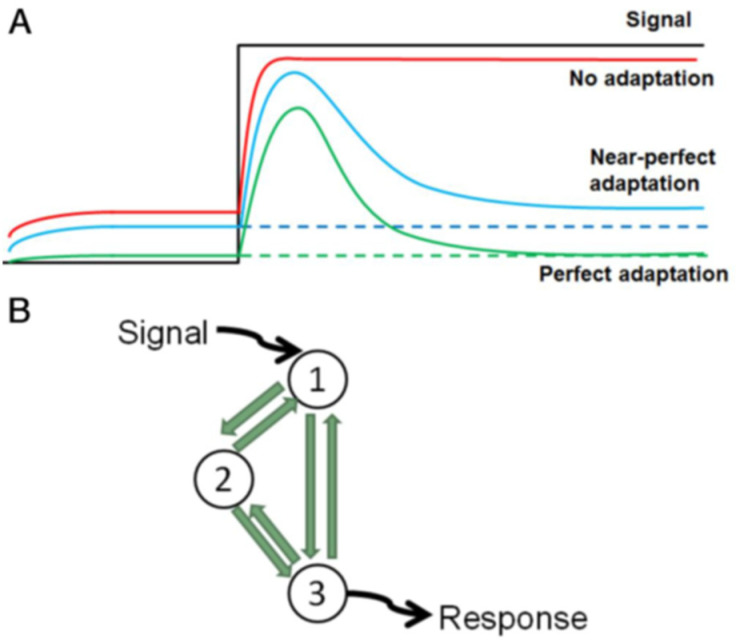
Basic concepts. (**A**) Perfect (green) and near-perfect (blue) adaptation in response to a persistent signal (black). (**B**) A three-node motif with six possible regulatory signals (green arrows). By specifying the interactions as + (activation), − (inhibition), or 0 (absent), we generate a universe of 729 distinct motif topologies.

**Figure 2 biology-12-00841-f002:**
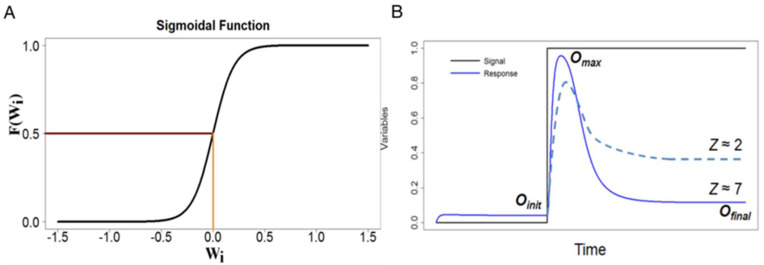
Sigmoidal response dynamics. (**A**) The sigmoidal function $F\left(W_{i}\right)=\frac{1}{1+e^{-\sigma W_{i}}}$ for *σ* = 10. (**B**) Two different responses (solid and dashed blue lines) to the signal (black), with their scores *Z* computed from Equation (4).

**Figure 3 biology-12-00841-f003:**
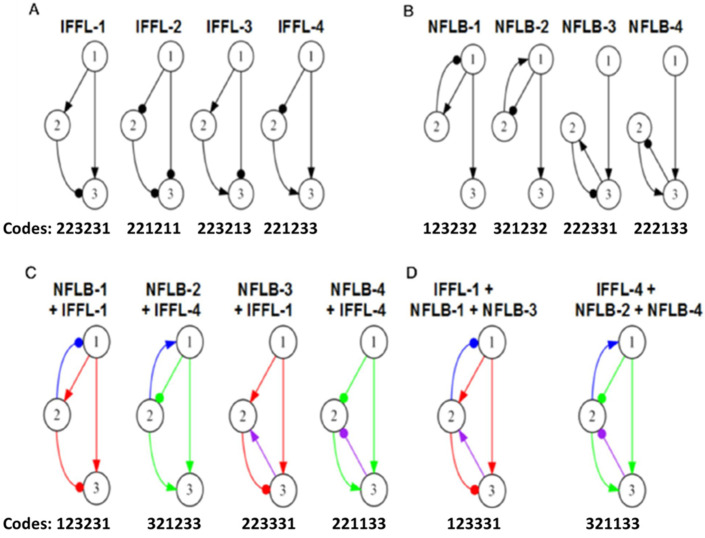
Common topologies supporting near-perfect adaptation. (**A**) Incoherent Feed Forward Loops (IFFLs). (**B**) Negative Feedback Loops with Buffering (NFLBs). (**C**) Topologies that combine the four NFLBs with the two dominant IFFLs. Red denotes IFFL-1, green denotes IFFL-4, blue denotes ‘upper’ NFLBs, purple denotes ‘lower’ NFLBs. (**D**) Topologies that combine each of the two high-scoring IFFLs with both an ‘upper’ and a ‘lower’ NFLB.

**Figure 4 biology-12-00841-f004:**
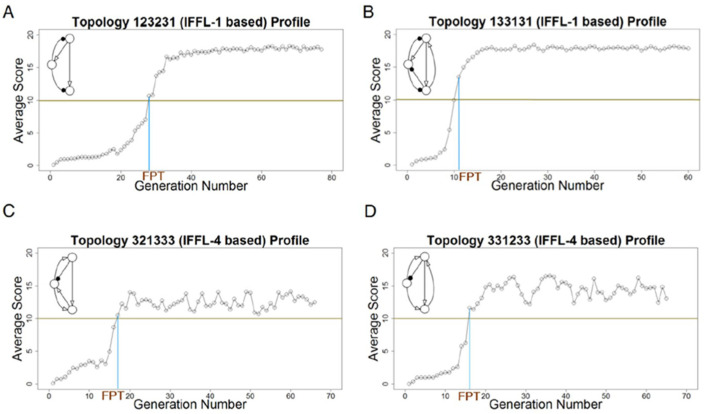
Examples of evolutionary simulations with micro-mutations only, showing average scores per generation with *N* = 20 and *R* = 20. (**A**,**B**) IFFL-1 based topologies. (**C**,**D**) IFFL-4 based topologies. FPT: First Passage Time.

**Figure 5 biology-12-00841-f005:**
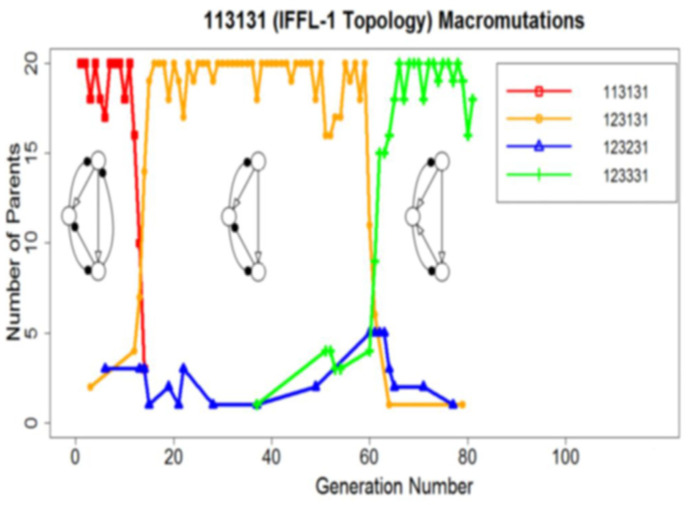
A sample run with macro-mutations, starting from an IFFL-1 topology.

**Figure 6 biology-12-00841-f006:**
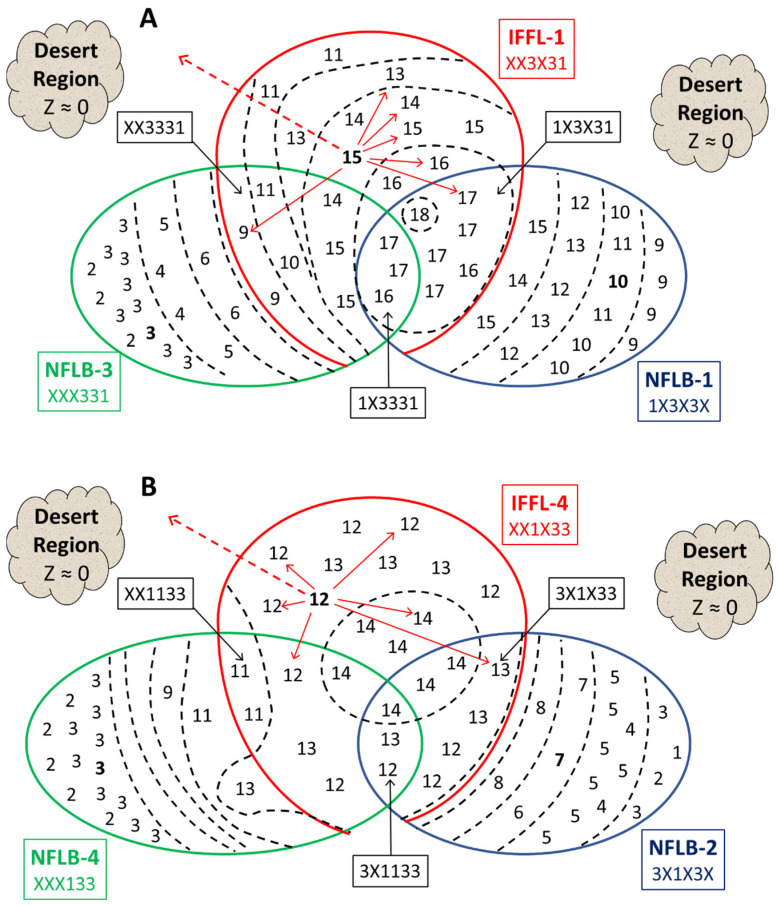
Venn diagrams illustrating the IFFL mesas in topology space. (**A**) M_IFFL1_ + M_NFLB1_ + M_NFLB3_. (**B**) M_IFFL4_ + M_NFLB2_ + M_NFLB4_. Each Venn diagram subsumes 63 separate topologies, whose fitness scores (from Table 3 and Table 4) are recorded in the relevant subset to which they belong. The scores are separated by dashed lines at 2 unit intervals. *Pure* motifs are indicated by bold scores. Macro-mutations of the *pure* IFFL-1 and -4 motifs are indicated by red lines. In each case, there are six solid red lines indicating macro-mutations that remain within M_IFFL1_ and M_IFFL4_, respectively, and the dashed red line represents the six macro-mutations that carry the topology into the ‘desert’ region of low fitness. Notice that no single macro-mutation can carry a topology from M_IFFL1_ to M_IFFL4_ or vice versa. Both M_IFFL1_ and M_IFFL4_ are evolutionarily stable with respect to macro-mutations that remain within the mesa. Macro-mutations in the desert region are likely to go extinct or to pick up a second mutation and enter M_IFFL1_ or M_IFFL4_.

**Figure 7 biology-12-00841-f007:**
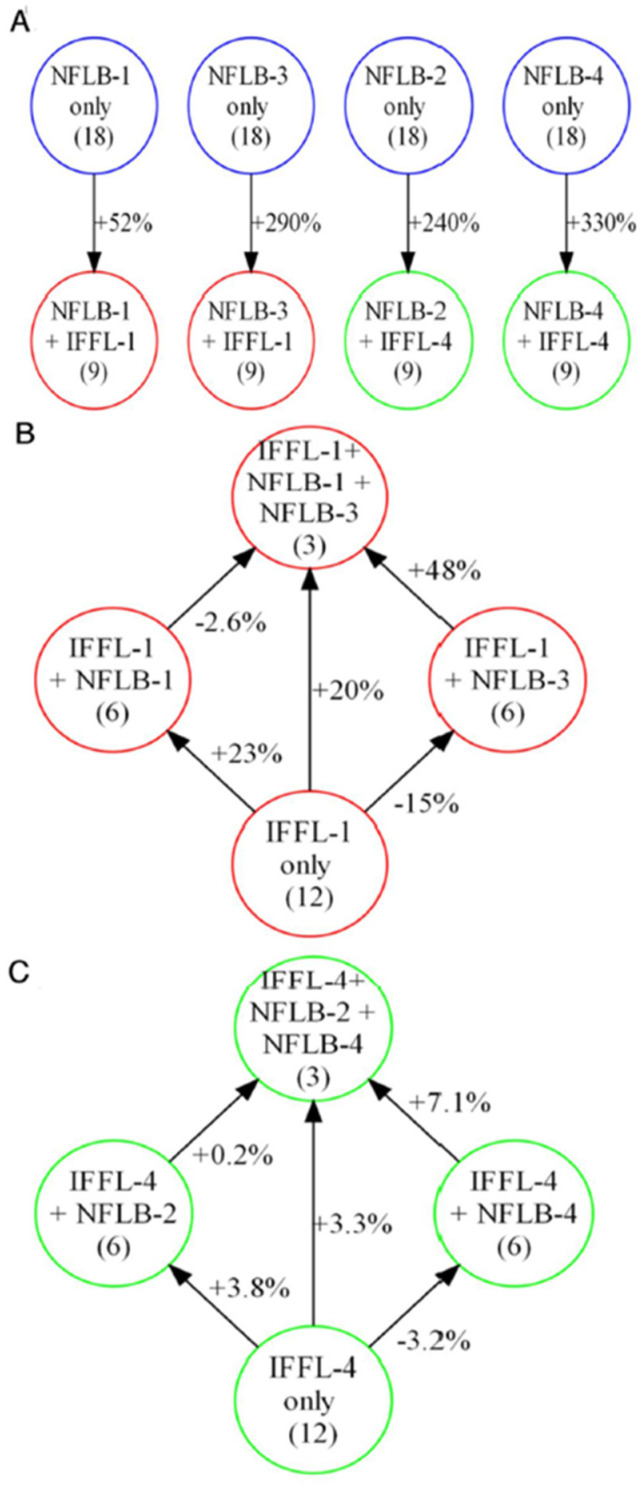
Fitness changes when NFLB and IFFL topologies are combined. (**A**) The fitness of an NFLB-only topology is increased by adding an IFFL-1 motif. (**B**) The fitness of an IFFL-1 only topology undergoes relatively small percentage changes (+ or −) when admixed with NFLB-1 and/or -3. (**C**) Similarly, for IFFL4-only admixed with NFLB-2 and/or -4.

**Figure 8 biology-12-00841-f008:**
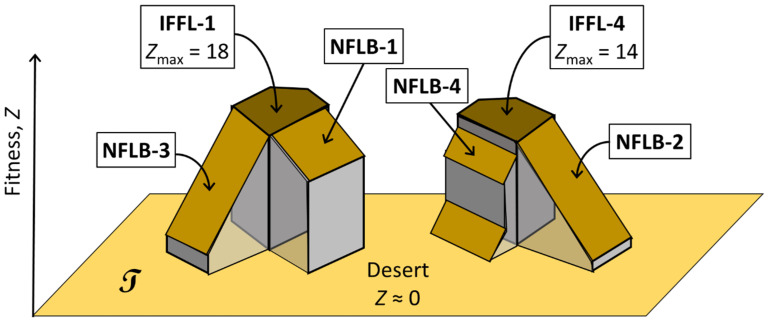
Fitness landscape in topology space. IFFL-1 and IFFL-4 form two evolutionarily stable subsets in topology space T.

**Table 1 biology-12-00841-t001:** The role and range of each parameter in the model.

Parameter	Role	Range
*ω_i_* _0_	Offsets	[−2, 2]
*ω_ij_*	Interaction Coefficients	[−1, −0.1] 0 [0.1, 1]
*γ*_1_, *γ*_2_	Rate constants	[0.1, 3]
*γ* _3_ ^−1^	Time scale	1
*σ*	Sigmoidicity	10

**Table 2 biology-12-00841-t002:** The highest scoring toplogies from an initial evolutionary simulation with both micro- and macro-mutations.

Code	< *Z* >	Code	< *Z* >	Code	< *Z* >
123331 (1,3)	14.18	113331 (1,3)	10.40	113231 (1)	7.81
133231 (1)	13.84	233231	10.24	233331 (3)	7.40
123231 (1)	13.46	121333	10.07	121233	7.26
133331 (1,3)	13.35	333331 (3)	9.19	213331 (3)	7.04
233131	13.31	333231	9.12	321233 (2)	6.83
321133 (2)	12.10	131233	9.06	323231	6.50
223131	11.42	223331 (3)	8.79	221333	6.41
123131 (1)	11.14	133131 (1)	8.78	333131	6.07
131133 (4)	11.11	**223231**	8.39		
111233	10.50	113131 (1)	8.08		

Red: IFFL-1 topologies (XX3X31); green: IFFL-4 topologies (XX1X33); purple: NFLB-1,2,3,4 topologies; < *Z* >: average score. Code 223231 (bold) is the only ‘minimal’ topology (IFFL-1).

**Table 3 biology-12-00841-t003:** The average scores < Z > and First Passage Times (FPT) of all IFFL-1 (XX3X31) and IFFL-4 (XX1X33) topologies.

(A) IFFL-1 Topologies			(B) IFFL-4 Topologies		
Code	< *Z* >	FPT	Code	< *Z* >	FPT
113131	16.87	118	111133	12.48	15
113231	16.26	27	111233	11.45	70
113331	16.10	20	111333	14.04	25
123131	16.92	17	121133	11.96	31
123231	17.15	28	121233	12.02	100
123331	16.76	16	121333	13.47	26
133131	17.68	11	131133	12.57	33
133231	17.03	24	131233	13.64	23
133331	16.77	22	131333	13.55	42
213131	11.43	34	211133	11.41	18
213231	13.19	49	211233	11.89	26
213331	10.93	38	211333	13.25	12
223131	15.18	7	221133	13.67	66
**223231**	15.25	26	**221233**	**11.96**	**30**
223331	9.34	17	221333	12.15	17
233131	13.88	53	231133	11.27	53
233231	14.38	69	231233	12.19	32
233331	14.99	10	231333	12.61	80
313131	13.14	28	311133	13.64	24
313231	14.73	6	311233	12.73	31
313331	10.33	134	311333	11.81	32
323131	10.80	48	321133	12.94	10
323231	15.62	5	321233	12.90	31
323331	14.91	25	321333	12.53	17
333131	15.49	21	331133	12.55	22
333231	14.40	8	331233	14.36	16
333331	9.19	41	331333	14.34	41

The *pure* topologies are **bold-face**. These micro-mutation only simulations were done with *N* = *R* = 20. (A) Red, IFFL-1 + NFLB-1 topologies (1X3X31); underlined, IFFL-1 + NFLB-3 topologies (XX3331). (B) Green, IFFL-4 + NFLB-4 topologies (1X1X33); underlined, IFFL-4 + NFLB-2 topologies (3X1X33).

**Table 4 biology-12-00841-t004:** The average scores < *Z* > of all four NFLB topologies.

(A) NFLB-1		(B) NFLB-2		(C) NFLB-3		(D) NFLB-4	
Code	< Z >	Code	< Z >	Code	< Z >	Code	< Z >
133131	17.68	331233	14.36	123331	16.77	221133	13.67
123231	17.15	331333	14.34	133331	16.69	311133	13.64
133231	17.03	311133	13.64	113331	16.11	133133	13.20
123131	16.92	321133	12.94	233331	14.99	321133	12.94
113131	16.87	321233	12.90	323331	14.91	131133	12.57
123331	16.76	331133	12.55	213331	10.93	331133	12.55
133331	16.77	321333	12.53	313331	10.33	111133	12.48
113231	16.26	311333	11.81	223331	9.34	121133	11.96
113331	16.10	311233	10.40	333331	9.19	211133	11.41
133132	15.40	331332	8.20	122331	5.67	231133	11.27
133232	15.00	331132	7.58	132331	5.67	123133	10.46
133332	14.29	321132	7.39	112331	5.05	113133	8.63
133133	13.20	**321232**	**7.10**	321331	4.55	322133	3.25
133233	12.48	311132	6.23	121331	4.19	332133	3.22
123332	12.29	321231	5.09	131331	3.91	312133	2.90
123132	12.06	331232	4.78	212331	3.39	112133	2.74
133333	11.99	331231	4.76	322331	3.36	122133	2.68
123333	10.71	311232	4.74	312331	3.35	**222133**	**2.64**
113332	10.69	331131	4.73	**222331**	**3.13**	212133	2.64
123233	10.55	311231	4.55	332331	3.11	323133	2.63
123133	10.46	321331	4.55	232331	2.96	313133	2.60
113232	10.32	321332	4.42	311331	2.80	132133	2.59
**123232**	**9.80**	311332	4.30	331331	2.80	233133	2.43
113132	9.35	331331	2.80	231331	2.77	223133	2.27
113333	8.97	311331	2.79	221331	2.49	213133	2.24
113233	8.73	311131	1.64	211331	2.40	232133	2.12
113133	8.63	321131	1.33	111331	2.13	333133	1.81

The *pure* topologies are **bold-face**. (A,C) Red, NFLB + IFFL-1 topologies; underlined, NFLB-1 + NFLB-3 topologies (1X331). (B,D) Green, NFLB + IFFL-4 topologies; underlined, NFLB-2 + NFLB-4 topologies (3X1133).

## Data Availability

No new data were created. Equations and simulations were codified in the R programming language by R.S.

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
