# Peer review of "Evolutionary Stability of Small Molecular Regulatory Networks That Exhibit Near-Perfect Adaptation"

_biology, 2023, doi:10.3390/biology12060841_

Round 1

Reviewer 1 Report

Singhania and Tyson discusses the evolutionary stability of 3-node networks that exhibit near-perfect adaptation using modeling.  The authors summarize the current state-of-art regarding near-perfect adaptation and use their search algorithm to find similar but not exactly the same set of mechanisms to achieve near-perfect adaptation.  The methods and results are well described and this study could be of interest to readers.

Still, there are too many equations and tables that hamper the understanding in some cases. It would be best to move some tables to supplement and show only the relevant parts of tables in the main text. 

Overall, the paper is well written. 

Other minor comment:

Line 172: the Wilson-Cowan equations similarities to Eq. -> the Wilson-Cowan equations have similarities to Eq. 

Author Response

Please review via the attachment.

Reviewer 2 Report

I find the manuscript by Tyson and Singhania quite appealing, despite the fact that I am not entirely familiar with the field. The investigation of adaptive response is certainly important, the work is well presented and deserves to be published.

I would be happy if the authors were to add comments on the possible role of noise and fluctuations in their systems.

Author Response

Please review via the attachment.

Reviewer 3 Report

Singhania and Tyson developed a mathematical model to study the evolutionary stability of small molecular regulatory networks that exhibit near-perfect adaptation. They used an evolutionary algorithm to search the parameter space and simulated the effect of mutations on the evolution of these networks. The manuscript is well written and the findings in this study is very interesting. Therefore, I recommend publishing the manuscript after a minor revision.

Minor comment:

The results and discussions have a weak connection to the biological implication. The authors should discuss more about the biological implication of their findings. For example, they found the topologies that have IFFLs are usually evolutionarily stable. Are there any well-known biological pathways containing this type of topology? If this type of topology is evolutionarily stable, I anticipate that they will be enriched in the essential pathways. Have the authors examined this?

Author Response

Please review via the attachment.
